# Impact of a Live Attenuated Classical Swine Fever Virus Introduced to Jeju Island, a CSF-Free Area

**DOI:** 10.3390/pathogens8040251

**Published:** 2019-11-20

**Authors:** SeEun Choe, Jae-Hoon Kim, Ki-Sun Kim, Sok Song, Wan-Choul Kang, Hyeon-Ju Kim, Gyu-Nam Park, Ra Mi Cha, In-Soo Cho, Bang-Hun Hyun, Bong-Kyun Park, Dong-Jun An

**Affiliations:** 1Viral Disease Division, Animal and Plant Quarantine Agency (APQA), Gimcheon, Gyeongbuk 39660, Korea; ivvi59@korea.kr (S.C.); kisunkim@korea.kr (K.-S.K.); ssoboro@naver.com (S.S.); changep0418@gmail.com (G.-N.P.); rami.cha01@korea.kr (R.M.C.); chois38@korea.kr (I.-S.C.); hyunbh@korea.kr (B.-H.H.); park026@korea.kr (B.-K.P.); 2College of Veterinary Medicine and Veterinary Medicine Institute, Jeju National University, Jeju Island 63243, Korea; kimjhoon@jejunu.ac.kr; 3Jeju Special Self-Governing Provincial Veterinary Research Institute, Jeju Island 63344, Korea; kwc1041@korea.kr (W.-C.K.); bluemouse@korea.kr (H.-J.K.); 4Colleage of Veterinary Medicine, Seoul University, Gwanak-ro, Gwanak-gu, Seoul 08826, Korea

**Keywords:** classical swine fever virus (CSFV), LOM vaccine strain, Jeju LOM strain, omega value, transmission

## Abstract

Here, we examine the effects of LOM(Low virulence of Miyagi) strains isolated from pigs (Jeju LOM strains) of Jeju Island, where vaccination with a live attenuated classical swine fever (CSF) LOM vaccine strain was stopped. The circulation of the Jeju LOM strains was mainly caused by a commercial swine erysipelas (*Erysipelothrix rhusiopathiae*) vaccine mixed with a LOM vaccine strain, which was inoculated into pregnant sows of 20 pig farms in 2014. The Jeju LOM strain was transmitted to 91 pig farms from 2015 to 2018. A histopathogenic investigation was performed for 25 farms among 111 farms affected by the Jeju LOM strain and revealed pigs infected with the Jeju LOM strain in combination with other pathogens, which resulted in the abortion of fetuses and mortality in suckling piglets. Histopathologic examination and immunohistochemical staining identified CSF-like lesions. Our results also confirm that the main transmission factor for the Jeju LOM strain circulation is the vehicles entering/exiting farms and slaughterhouses. Probability estimates of transmission between cohabiting pigs and pigs harboring the Jeju LOM strain JJ16LOM-YJK08 revealed that immunocompromised pigs showed horizontal transmission (r = 1.22). In a full genome analysis, we did not find genetic mutation on the site that is known to relate to pathogenicity between Jeju LOM strains (2014–2018) and the commercial LOM vaccine strain. However, we were not able to determine whether the Jeju LOM strain (2014–2018) is genetically the same virus as those of the commercial LOM vaccine due to several genetic variations in structure and non-structure proteins. Therefore, further studies are needed to evaluate the pathogenicity of the Jeju LOM strain in pregnant sow and SPF pigs and to clarify the characteristics of Jeju LOM and commercial LOM vaccine strains.

## 1. Introduction

Classical swine fever virus (CSFV), a member of the genus *Pestivirus* within the family *Flaviviridae*, comprises a single positive-stranded RNA genome of approximately 12.3 kb, which encodes a polyprotein of 3898 amino acids [1]. The viral genome comprises a 5’ untranslated region (5’ UTR), an N-terminal protease (N^pro^), a capsid (C) protein, envelope (E) proteins (E^rns^, E1, E2, p7), non-structural (NS) proteins (NS2, NS3, NS4A, NS4B, NS5A, NS5B), and a 3’ UTR [2]. CSFV is categorized into three genotypes, 1, 2, and 3, each of which can be subdivided into three subgenotypes (1.1–1.3, 2.1–2.3, and 3.1–3.4) [3]. Classical swine fever (CSF) is a highly contagious multisystemic hemorrhagic viral disease of domestic pigs and wild boar, which can manifest as acute, subacute, chronic, or late onset disease [4]. Vaccination is used to prevent and/or reduce the number of outbreaks of CSF and, together with other control measures, was an important factor in eradicating CSF from Holland in 1985 and from the Rivas region of Nicaragua [5]. Historically, CSF vaccines have been based on attenuated strains, i.e., lapinized Chinese and tissue culture-adapted strains. Modified live vaccines (MLVs) based on several attenuated virus strains (e.g., C-strain, Thiverval, PAV-250, GPE-, and K-strains) are used most widely. The advantages and disadvantages of MLV vaccination during an outbreak have been described in previous studies [5]. Briefly, the advantages include ease of use, low cost, and an induction of life-long immunity by a single dose. However, in the 1980s and early 1990s (the time when a lapinized Chinese strain was used in Mexico), vaccinated pigs occasionally exhibited adverse reactions; some even died [5]. The CSF vaccine virus multiplies in actively replicating cells such as fetal cells and reticuloendothelial cells; some reports show that animals vaccinated with a Chinese strain virus had more respiratory infections due to a concurrent *Pasteurella multocida* infection [6]. In addition, the CSF vaccine virus may induce embryonic death and myoclonia congenita when administered to pregnant sows [5]. When herds that include pregnant sows are vaccinated for the first time, there can be a fall in sow fertility in the 6 weeks following vaccination, after which it returns to normal; therefore, researchers concluded that vaccination reduces herd productivity [5]. The OIE requirements for the safety of MLVs in young animals states that vaccination should not induce a high body temperature or leukopenia and should not allow horizontal transmission. For pregnant sows, safe MLV vaccination requires no transplacental transmission and no evidence of reversion-to-virulence after passage in piglets. The CSF MLV (LOM vaccine strain) used to vaccinate pigs in South Korea since 1974 has caused abortion in some pregnant sows [7], and prolonged virus shedding by immunosuppressed pigs after vaccination [8].

The South Korean government maintained a CSF (LOM) vaccine policy to control the disease on the mainland; however, vaccination on Jeju Island (located at the southern end of Korea) was stopped in 2000 (Jeju Island declared itself a “CSF-free region”). However, pig farms of Jeju Island have suffered continuous outbreaks of the LOM vaccine strain via various routes from the mainland, raising suspicion about the safety and reversion-to-pathogenicity of commercial LOM vaccine strains. Five outbreaks of the LOM vaccine strain infection of pigs have occurred over 19 years. The first (from 2004 to 2007) occurred through feed contaminated with the commercial LOM vaccine strain present in an animal plasma protein supplement [9]. The second was caused by the delivery of incorrect vaccine material from the mainland; in 2010, one farm accidently inoculated pigs with a vaccine mixed with the LOM vaccine strain. The third and fourth occurred on one farm (2012) and two farms (2013); the route of LOM vaccine strain exposure was not revealed, but feed contaminated with the LOM vaccine strain, or an unintentional injection of an incorrect CSF (LOM) vaccine from the mainland was suspected. In the fifth case (in 2014), pregnant sows on 20 pig farms were inoculated with a commercial swine erysipelas vaccine mixed with the LOM vaccine strain. Vaccination was stopped immediately, but a total of 111 pig farms (20 in 2014, 22 in 2015, 32 in 2016, 26 in 2017, and 11 in 2018) were exposed to the Jeju LOM strains (LOM strains isolated from Jeju pigs). Farm-to-farm transmission patterns showed mainly between high-density pig farms in the Hanlim region (located in northwestern Jeju) from 2015 to 2018 and the Daejeong region (located in southwestern Jeju) from 2016 to 2018.

Here, we investigated pig-to-pig and/or farm-to-farm transmissions, and the possibility of reversion-to-pathogenicity by comparing genetic mutations in the Jeju LOM strain viruses.

## 2. Results

### 2.1. Histopathological Analysis to Detect Infection by CSFV (Jeju LOM) Alone or Co-Infection with CSFV (Jeju LOM) and Other Pathogens

Overall, 122 samples (from 100 piglets and 22 fetuses) obtained from 25 pig farms were tested; all were positive for CSFV (Jeju LOM). The Jeju LOM strain was identified in 103 of 122 samples (81 piglets and 22 fetuses). Antigens derived from the Jeju LOM strain were detected in tissue samples from 51 suckling piglets, 14 piglets were infected with Jeju LOM alone, and the remaining 37 were co-infected with Jeju LOM plus enteric pathogens (six with *Clostridium* spp, six with *E. coli*, three with PEDV, and four with rotavirus), respiratory pathogens (three with PRRSV), or *Streptococcus* (n = 7), and *Staphylococcus* spp. (n = 2) (Table 1). Samples from the 37 suckling piglets co-infected with the Jeju LOM strain and other pathogens showed evidence of interior visceral hemorrhage (18 of the kidney, 12 of the exo-endocardium, and nine of the lung), and 14 had non-purulent brain lesions (perivascular cuffing, gliosis, and neuronophagia). Fourteen piglets infected with the Jeju LOM strain only showed interior visceral hemorrhage (eight of the kidney, six of the exo-endocardium, and three of the lung), and six had non-purulent brain lesions (perivascular cuffing, gliosis, and neuronophagia). Seven of the 14 suckling piglets identified to have CSF-like specific histopathologic lesions (Figure 1). Pathogenic lesions in weaning pigs included broncho-interstitial pneumonia or fibrinous lobor pneumonia (n = 16), lung hemorrhage (n = 10), kidney hemorrhage (n = 9), peripheral lymph node hemorrhage (n = 11), exo-endocardium hemorrhage (n = 5), and non-suppurative encephalitis of brain and spinal cord (n = 5) (Table 1). Co-infection of 22 fetuses with other pathogens (i.e., PPV, ADV, EMCV, JEV, PRRSV, and PCV2) was not confirmed, and no specific pathogenic lesions were observed in their organs. Organ tissue immunohistochemistry (IHC) staining detected Jeju LOM strains in 25 of 48 suckling pigs; 17 were cases of co-infection and eight were single infections. The following organs harbored the Jeju LOM strain: tonsil (40%), spleen (22.9%), lymph node (15%), lung (14.6%), small intestine (4.3%), kidney (4.2%), and liver (2.1%) (Table 2). IHC staining of pathogenic tissues from weaning pigs detected the Jeju LOM strain in internal organs (28.6%; 6/21): lymph nodes (20%), spleen (16.7%), lung (15.0%), and tonsil (5%) (Table 2). However, no virus was detected in other internal organs (heart, liver, intestine, spinal, and brain). One of the 22 fetuses showed an infection of the kidney alone (Table 2).

### 2.2. Detection of Antibodies on Pig Farms Exposed to the Jeju LOM Strain

The average anti-CSF (Jeju LOM) antibody-positive rates on seven pig farms in the Jeju region was as follows: 87.1% ± 4.2% of sows, 77.8% ± 8.6% of suckling piglets, 24.2% ± 14.1% of weaning pigs, 21.4% ± 11.2% of growing pigs, and 38.5% ± 13.7% of finishing pigs (Table 3). There were significant differences (*p* < 0.05) in the average anti-CSF (Jeju LOM) antibody-positive rates between sows and weaning pigs/growing pigs, and between suckling piglets and growing pigs, respectively. Serum neutralizing antibody (log_2_) titers in pigs on antibody-positive farms were 9.42 ± 0.25 (log_2_) in sows, 7.04 ± 0.54 (log_2_) in suckling pigs, 7.2 ± 1.04 (log_2_) in weaning pigs, 5.52 ± 0.12 (log_2_) in growing pigs, and 7.44 ± 0.74 (log_2_) in finishing pigs (Table 3). There were significant differences in the average anti-CSF (Jeju LOM) neutralizing antibody titers between sows and growing pigs (*p* < 0.05).

### 2.3. Farm-Slaughterhouse-Farm Transmission

Of the 242 samples collected from vehicles at a slaughterhouse in Jeju, 151 (62.4%) were positive for Jeju LOM antigens by qRT-PCR. The detailed results regarding infected sites were as follows: 66.2% (47/71) driver foot floor, 38% (27/71) vehicle wheels, 71.6% (53/74) pig-holding compartment, and 92.3% (24/26) “other”. Among the virus-positive samples, 92 positive samples had an estimated TCID_50_/mL of 10^3.0–3.9,^ and 11 samples had an estimated TCID_50_/mL of 10^4.0–4.9^. One sample had a TCID_50_/mL > 10^5.0^ (Table 4). A peroxidase-link immunosorbent assay (PLA) virus viability test revealed strong staining of 13 samples (4 driver foot floor, 2 vehicle wheels, and 7 pig-holding compartment) and weak staining of 32 samples; 129 samples were unconfirmed due to cellular contamination by bacteria (Table 4).

### 2.4. Pig-to-Pig Transmission and Reproduction Rate (R)

Six pigs were inoculated with the Jeju LOM strain JJ16LOM-YJK08 and, after 24 h, were placed in a pen with six non-inoculated pigs. For Group 1, we used pigs (unhealthy) with PRRSV or PCV2, and Group 2 used pigs (healthy) without specific wasting diseases. In Group 1, two non-inoculated pigs were positive for anti-CSF (Jeju LOM) antibodies on Day 21 and Day 28. However, no non-inoculated pigs in Group 2 had detectable anti-CSF (Jeju LOM) antibodies on Day 45 (Table 5). The transmission possibility estimate for group 1 was R0 = 1.22 (95% confidence interval (CI), 0.980–1.765), whereas that for Group 2 was R0 = 0.00 (95% CI, not applicable) (Table 5).

### 2.5. Comparison of LOM Strain Genome Sequences

The amino acid sequences of four commercial LOM vaccine strains were compared with those of five Jeju LOM strains (2004–2007), and three unique amino acid changes in the E1 (V-577-A/M) and NS4B (M-2378L and V-2383A) proteins were detected (Appendix A). Comparison of the Jeju LOM strain JJ04LOM-Tamra01 (2004) with the four other Jeju LOM strains (2005–2007) revealed six amino acid changes: E^rns^ (D-386-N, R-480-G), E2 (L-1065-S), NS3 (K-1165-R), NS4B (A-2352-V), and NS5A (N-2816-T) (Appendix A). Comparison of the four commercial LOM vaccine strains with 12 Jeju LOM strains (2014–2018) identified unique changes in the N^pro^ (K-57-R, L-143-Q), E^rns^ (Y-351-H, R-476-S), E1 (I-651-T), NS3 (V-1381-I, H-1584-N, K-2006-I), NS4B (M-2348-I, T-2371-I, I-2398-M, and V-2483-A), NS5A (A-2978-T), and NS5B (N-3409-S and S-3786-N) proteins (Appendix A).

### 2.6. Root-to-Top Divergence and Positive Selection Analyses

Application of the heuristic residual mean squared method to all strains (the commercial LOM vaccine and Jeju LOM strains) using the TempEst program revealed a slope of 3.93 × 10^−5^ (rate), an X-intercept (TMRCA) of 1797.24, a correlation coefficient of 0.1272, an R squared value of 1.6173 × 10^−2^, and a residual mean squared value of 6.9076 × 10^−6^. Root-to-top divergence for Jeju LOM strain JJ07LOM-JSG02 was >0.0130. Mainly, the Jeju LOM strains isolated from the field showed a high divergence value of between 0.0070 and 0.0130 (Figure 2). However, seven LOM vaccine strains (excluding 88LOM-Suri) and two Jeju LOM strains (JJ14LOM-WSH01 and JJ17LOM-LHH10) showed a low divergence value (0.0040–0.0060) (Figure 2). The omega value (*dN/dS*) for the commercial LOM vaccine strains and the Jeju LOM strains isolated from the field (2004–2007 and 2014–2018) showed high homology with respect to the E^rns^, E1, E2, NS2, NS3, NS4A, NS4B, NS5A, and NS5B proteins (Figure 3A). However, the omega value for the C and P7 proteins of commercial LOM vaccine strains was a little higher (0.34224 and 0.43781, respectively) than that of the five Jeju LOM strains isolated in 2004 to 2007. The omega values for the C, E1, and E2 structural proteins of the Jeju LOM strains isolated in the field in 2014 to 2018 were higher than for the NS proteins (Figure 3A). Jeju LOM strains isolated from pigs from 2004 to 2007 were 98.8% to 99.2% identical at the nucleotide level and 99.2% to 99.4% identical at the amino acid level, with an omega value of 0.14088. Jeju LOM strains isolated from 2014 to 2018 showed 98.2% to 99.4% identity at the nucleotide level and 99.1% to 99.6% identity at the amino acid level, with an omega value of 0.16196 (Appendix A). The Bayes empirical Bayes (BEB) analysis of Jeju LOM strains isolated from 2004 to 2007 showed the possible inclusion mutation sites at 237, 259, 577, and 2467 aa positions, but a BEB analysis of Jeju LOM strains from 2014 to 2018 showed mutations at 173, 176, 386, 564, 1337, 2676, 2988, and 3605 aa positions. A native empirical Bayers (NEB) analysis performed with Jeju LOM strains (2014–2018) revealed the mutation site of the 564 aa position (*p* > 99%; Appendix A).

### 2.7. Geographic Distance and MCC Tree Analysis

A correlation between geographic distance and genetic *p*-distance of the Jeju LOM strains contaminating the commercial LOM vaccine strain 16LOM-KM00 from 2014 to 2018 was confirmed, as shown in Figure 3B. Jeju LOM strain JJ14LOM-WSH01, isolated at a pig farm in the Aeweal region in 2014, showed a genetic *p*-distance from commercial LOM vaccine strain 16LOM-KM00 of 0.0018. Eleven Jeju LOM strains (Hanlim region: geographic distance, 13.5–20.1 km) and four Jeju LOM strains (Daejeong region: geographic distance, 27.4–28.1 km) isolated from pig farms showed a genetic *p*-distance of 0.0045–0.0139 and 0.0011–0.0086, respectively (Figure 3B and Figure 4). Among genotypes 1, 2, and 3 in the beast tree constructed after global analysis of the complete E2 protein of CSFV, all LOM strains (including the Jeju LOM strains) belonged to independent groups within subgenotype 1.1 (Figure 5). From the mid-1980s, LOM strains were divided into two clusters: a lower cluster comprising commercial LOM vaccine strains and an upper cluster comprising mainly Jeju LOM strains isolated from pigs (Figure 5). In the above cluster, Jeju LOM strains isolated from pigs on Jeju Island were divided according to the year of isolation (2004–2007 and 2014–2018; Figure 5). The mean tMRCA for the LOM strains was 41.466, with an ESS (effective sample size) of 2340.2436 and a 95% highest posterior density (HPD) interval of 39.018–44.3142. The clock rate for LOM strains was 5.215 × 10^−4^, with a 95% HPD interval of 4.1721 × 10^−4^–6.159 × 10^−4^.

## 3. Discussion

The main characteristics of the LOM strain are attenuated; however, inoculation into pregnant sows lacking anti-CSF antibodies triggers abortion [7]. In addition, LOM vaccination of immunocompetent pigs leads to an antigen release period of more than 4 weeks, which results in the worsening of lesions if an animal is co-infected with other pathogens [8]. A previous study tested the effect of a live attenuated BVDV vaccine on pig-to-pig transmission and found that the R value was <1 [10]. Here, we found that transmission of the Jeju LOM strain differed significantly depending on the health status of the pig (Group 1 (unhealthy): r = 1.22; and Group 2 (healthy): r = 0). This is because, as for other wasting diseases, the period of virus release from infected pigs is prolonged and the amount of virus shed in the feces increases [8]. A commercial LOM vaccine strain showed 10% of the average transmission rate from inoculated pigs to cohabitating non-inoculated pigs [11]. Although a small number of pigs were used for the transmission experiment in this study, the transmission characteristics of the Jeju LOM strain were revealed to be similar to the commercial LOM vaccine. In a previous study, after oral inoculation of the LOM vaccine strain of 2 × 10^4.0^TCID_50_/mL, the LOM vaccine virus was detected in blood and feces in the 9 days post-inoculation [9]. We also detected 11 samples of live Jeju LOM viruses with more than 10^4.0^TCID_50_/mL in the slaughterhouse experiment, which support the possibility of farm-slaughterhouse-farm transmission.

From 2004 to 2005, LOM-infected pigs in Jeju showed varying symptoms. The clinical symptoms showed both the similarities and differences from those associated with CSF [10]. Similarities to more current Jeju LOM strain infections (2014–2018) included skin ulcers, ulceration of the tonsil, petechial hemorrhage in the kidney, button ulceration of the ileocecal region, infarction of the spleen, and pneumonia. Differences in histologic findings were a weakness in endothelial cell lesions of organs such as the spleen, lymph nodes, and brain. These lesions seem to be more related with PRRS not CSF. In addition, meningitis was observed in the central nervous system, CSF did not show meningitis, and lymphoid organs showed little increase in volume other than lymphocytic atrophy. The Jeju LOM strain that was prevalent from 2004 to 2007 was not related to CSF pathogenicity [10]. IHC (2004–2005) was Jeju LOM virus positive in 52.1% (25/48) of suckling piglets and 28.6% (6/21) of weaned pigs [10]. Our Jeju LOM antigen results (2014–2018) were lower than the results of the Jeju LOM antigen (2004–2005) test (58.3%) from a previous report [10]. Jeju LOM strains (2014–2018) were also suspected to have caused CSF-like lesions in 7 young piglets (average age of 10 days), which was proved by histopathological findings and IHC analysis. It may be deduced that some CSF lesions in suckling piglets, as well as abortion during pregnancy, are associated with LOM strains. However, in this case, it is unlikely that persistent infection will continue to circulate the virus among pigs.

Some experts suggest that Jeju LOM strains detected on Jeju pig farms from 2014 to 2018 have recovered pathogenicity and that it is now different from that of the commercial LOM vaccine strain on the current market. However, gene analysis of Jeju LOM strains revealed no evidence for restoration of pathogenicity. Previous studies have identified differences in pathogenic regions between the ALD and GPE strains as E2 (T830A), NS4B (V2475A, A2563V), and N^pro^ (N136D) [12,13]. Other studies revealed that Shimen’s pathogenicity-related gene locus is within E2 (T745I, M979K) [14] and E2 (N850S) [15]. Our analysis of the amino acid sequences of the commercial LOM vaccine strain and the Jeju LOM strains detected in Jeju pigs revealed no changes at these sites. In addition, although there are differences in pathogenicity linked to changes in N-linked glycosylation sites on the E2 protein of CSF, there was no change of N-linked glycosylation sites in the Jeju LOM strains [16]. A study comparing the complete nucleotide and amino acid sequences of the ALD and GPE strains revealed 98.2% and 98.8% identity, respectively [17]. Although the Jeju LOM strain (2014–2018) harbors several amino acid mutations, it is difficult to interpret this as conferring recovery of pathogenicity or additional attenuation. There was no significant difference in the gene mutation rate among Jeju LOM strains, nor was there a significant difference in omega values. Therefore, Jeju LOM strains (2014–2018) from Jeju cannot be considered a pathogenically-reversed LOM strain, which is virulent in pigs but shows a safety profile characteristic of the commercial LOM vaccine strain described in previous study [7]. More precisely, the presence of Jeju LOM strains with recovered pathogenicity should be evaluated in animals (i.e., pregnancy sows and SPF pigs).

In conclusion, the CSF-like histopathogenic lesions of Jeju pigs revealed to be more related to other viral pathogens rather than the Jeju LOM strains (2014–2018), despite the presence of the Jeju LOM strain in organs of piglets. We also confirmed that the pig-to-pig transmission of the Jeju LOM strain and farm-to-farm transmission may have been caused by vehicles visiting the slaughterhouse. Although we found some genetic differences between the Jeju LOM strains (2014–2018) and commercial LOM vaccine strains, more pathogenesis studies may be needed using animals such as pregnant sows and SPF pigs.

## 4. Materials and Methods

### 4.1. Detection of Histopathologic Lesions and Immunohistochemical Straining Analysis

From 2014 to 2018, tests in this study performed by the diagnostic services of a veterinary medicine college in Jeju national university to detect animal infectious disease identified 122 samples (100 piglets and 22 fetuses) from 25 pig farms that were positive for CSFV by RT-PCR, histopathology, and IHC. The amplification targets for RT-PCR were CSFV [9], porcine circovirus type 2 (PCV2) [18], porcine reproductive and respiratory syndrome virus (PRRSV), swine influenza virus (SIV), cytomegalovirus (PCMV), *Salmonella* spp, *Streptococcus* spp, *Actinobacillus, pleuropneumoniae*, and *Haemophilus parasuis.* Histopathologic lesions specific to CSFV were examined by staining tissues with hematoxylin and eosin. The following were examined: skin (hemorrhage and endothelial damage), lymph nodes (peripheral hemorrhage, lymphoid depletion, and reticular cell hyperplasia), kidney and bladder (hemorrhage, swelling and peripheral hemorrhage), spleen (hemorrhagic infarction, endothelial damage), cecum and colon (necrotic or ulcerative colitis, vascular congestion, and sub-serous hemorrhage), heart (myocardial hemorrhage), and brain and spinal cord (nonsuppurative encephalitis, proliferation of endothelial cells, perivascular cuffing, microgliosis, and focal necrosis). To confirm the presence of CSF antigen in tissues, IHC was performed using the EnVision^TM^ peroxidase-conjugated polymer reagent (DAKO, Denmark). In brief, tissues were reacted with a primary antibody (mouse anti-CSFV) (WH303, Animal and Plant Health Agency, New Haw, Addlestone, UK), followed by EnVision^TM^/HRP, rabbit/mouse (EVN) regent (DAKO, USA), and 3,3'-diamino-benzidine tetrahydrochloride (DAB). The presence of CSF antigen is denoted by a dark brown deposit in the tissue section.

### 4.2. Detection of Anti-CSF Antibodies

Seven pig farms with the Jeju LOM infection were selected to estimate how Jeju LOM strains were introduced into the farms and how they spread within the farms. The anti-CSF antibody-positive rates for each pig farm and antibody titer levels for pigs within farms were examined according to age. Blood samples were collected from 20 pigs in each group: sows, suckling pigs (10–20 days old), weaning pigs (40–60 days old), growing pigs (90–120 days old), and finishing pigs (150–180 days old). To detect CSF-specific neutralizing antibodies, a neutralizing peroxidase-linked assay (NPLA) was performed according to the standards manual of the World Organization for Animal Health [19]. For PK-15 cell staining, the monoclonal antibody 3B6 (Median Diagnostics, Chuncheon, Korea) was used to detect the CSF E2 protein.

### 4.3. Environmental Samples Taken from a Slaughterhouse

In 2017, 242 samples were obtained from a slaughterhouse in Jeju to investigate CSFV (Jeju LOM strain). These included 71 samples from the driver’s foot floor, 71 from vehicle wheels, 74 from the vehicle’s pig-holding compartment, and 26 from other sites. Real-time quantitative PCR (qRT-PCR) was performed to detect the CSF antigen copy number in fecal and environmental samples. The VDx^®^ CSFV qRT-PCR (MEDIAN Diagnostic Co. Cat No. NS-CSF-31, Gangwon-do, Korea), which uses TaqMan probes, detects the CSFV 5' UTR with high specificity; it does not detect BVDV or border disease virus, which also belong to the *Pestivirus* genus. Briefly, the qRT-PCR program comprised the following steps: cDNA synthesis (50 °C, 30 min) and initial inactivation (95 °C, 15 min), followed by a two-step PCR comprising 42 cycles of denaturation (95 °C, 10 s) and extension (60 °C, 60 s). A peroxidase-linked immunosorbent assay (PLA) was used to confirm viability of Jeju LOM strains. Briefly, PK-15 cells (grown to 80% confluence in 24-well plates) were inoculated for 72 h at 37 °C with 10% tissue homogenates including the Jeju LOM strain in a minimum essential medium. The PK-15 cells were fixed in pre-chilled 80% acetone after 72 h and reacted with a 3B6 monoclonal antibody specific for CSFV E2. Subsequently, the PK-15 cells were reacted with biotinylated anti-mouse IgG (H + L) (VECTOR Laboratories, Cat No. BA-9200, Burlingame, CA, USA) and ABC solution (VECTOR Laboratories, Cat No. PK-4000, Burlingame, CA, USA). After staining with DAB peroxidase substrate (VECTOR Laboratories, Cat No. SK-4100, Burlingame, CA, USA) following the manufacturer’s instruction, the PK-15 cells were observed under a microscope.

### 4.4. Horizontal Transmission between Pigs

To investigate the transmission probability of the Jeju LOM strain among pigs, 40-day-old CSF antigen- and antibody-negative pigs from two pig farms (n = 24, 12 per farm) were used. Group 1 comprised of pigs with a wasting disease (PRRSV or PCV2) and Group 2 comprised of pigs with specific non-disease. Six pigs from each group were inoculated intramuscularly with 2 mL (10^4.0^TCID_50_/mL) of the Jeju LOM strain JJ16LOM-YJK08, which is a representative strain circulating in Jeju between 2014 and 2018. After 24 h, the six pigs from each group were housed with non-injected pigs and observed for clinical signs and symptoms. Blood was collected every week for 45 days and tested in the NPLA assay and by qRT-PCR. The estimated reproduction number (R0) and confidence interval (CI) were calculated as described previously [20] using the following formula: R0 = −In((1−AR)/S0)/(AR−(1−S0)); CI = AR ± 1.96$\sqrt{\mathrm{ARx}\left(1-\mathrm{AR}\right)/n}$).

### 4.5. Genomic Analysis of Commercial LOM Vaccine Strains and Jeju LOM Strains

The complete genome sequences of seven commercial LOM vaccine strains (16LOM-GC00, 16LOM-JY00, 16LOM-KM00 and 16LOM-KR00 (collected from a market in 2016), 02LOM-JY00 (collected from a market in 2002), 88LOM-suri (isolated from LOM-850 strain in 1988), and LOM-850 (original master seed in 1987)) were examined. In addition, five Jeju LOM strains isolated from pigs on Jeju Island between 2004 and 2007, and 16 Jeju LOM strains isolated from pigs on Jeju Island between 2014 and 2018, were analyzed. Positive selection analysis of the complete ORF (N^pro^, C, E^rns^, E1, E2, P7, NS2, NS3, NS4A, NS4B, NS5A, and NS5B) was conducted using several models available in the BASEML and CODEML modules of the PAML 4.6 software package [21]. Different values of the non-synonymous/synonymous (*dN/dS*) ratio (the omega parameter), were considered in accordance with the user manual. A *dN/dS* ratio of <1 indicates a purifying selection, a *dN/dS* ratio = 1 suggests an absence of selection (i.e., neutral evolution), and *dN/dS* >1 indicates a positive selection. The Bayes empirical Bayes (BEB) calculation of the posterior probabilities of site classes was used to calculate the probability that a site is under positive selection pressure [22]. TempEst (formerly known as “Path-O-gen”) is a tool for investigating the temporal signals and “clocklikeness” of molecular phylogenies. The contemporaneous trees (in which all sequences were collected at the same time) and dated-tip trees (in which sequences are collected at different dates) were analyzed using the TempEst v1.5.1 program, which is temporal signal estimator tool [23]. The genome sequences of the Jeju LOM strains (n = 21) and commercial LOM vaccine strains (n = 8) in this study were submitted to the GenBank under accession numbers MN558862–MN558889.

### 4.6. Maximum Clade Credibility Tree

Complete sequences for the E2 gene of 145 CSFV isolates available in the GenBank Database, including complete E2 gene sequences of 8 commercial LOM vaccine strains and 21 Jeju LOM strains, were used to generate a BEAST input file using BEAUti within the BEAST package v1.8.1 [24]. Rates of nucleotide substitution per site and per year, and the most recent common ancestor (tMRCA), were estimated using a Bayesian MCMC approach. Each dataset was simulated with the following options: generation = 100,000,000, burn-in of 10%, and ESSs > 200. The exponential clock and expansion growth population model in the BEAST program was used to obtain the best-fit evolutionary model. The resulting convergence was analyzed using Tracer 1.5 [25]. Trees were summarized as a maximum clade credibility (MCC) tree using TreeAnnotator 1.7.4 [26] and visualized using Figtree 1.4 [27]. For each tree node, estimated divergence times and 95% HPD intervals, which summarize statistical uncertainties, were indicated.

### 4.7. Statistical Analysis

All statistical analyses were performed using the GraphPad Prism software, version 6.0, for Windows. Data were analyzed using one-way analysis of variance, followed by Tukey’s multiple-comparison test. Groups showing significant differences (*p* < 0.05) at the same time point are indicated by different letters.

## Figures and Tables

**Figure 1 pathogens-08-00251-f001:**
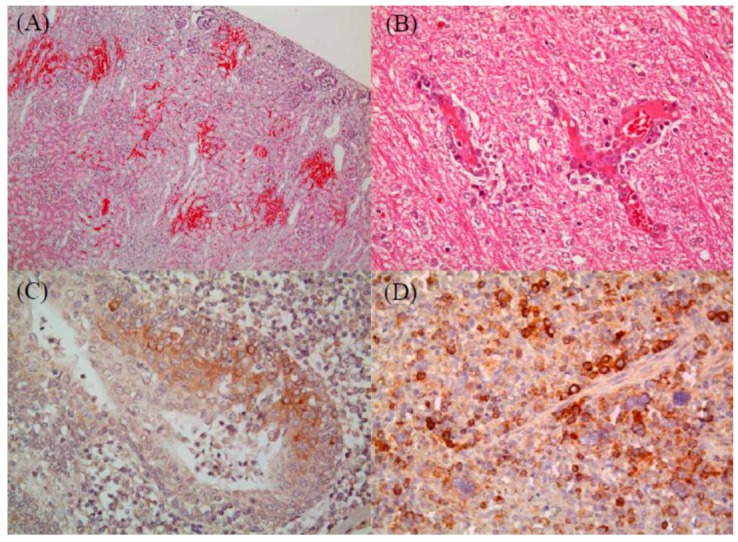
Immunohistochemistry (IHC) staining to detect histopathogenic lesions. Severe multifocal hemorrhages in the renal cortex of a suckling piglet (hematoxylin & eosin (HE); mag ×100) (**A**). Perivascular cuffing in the white matter of the cerebellum of a suckling piglet (HE; mag ×400) (**B**). Brown-stained viral antigens in the cryptal epithelium of the tonsil (IHC; mag ×400) (**C**). Brown-stained viral antigens in macrophages infiltrating the spleen (IHC; mag ×400) (**D**).

**Figure 2 pathogens-08-00251-f002:**
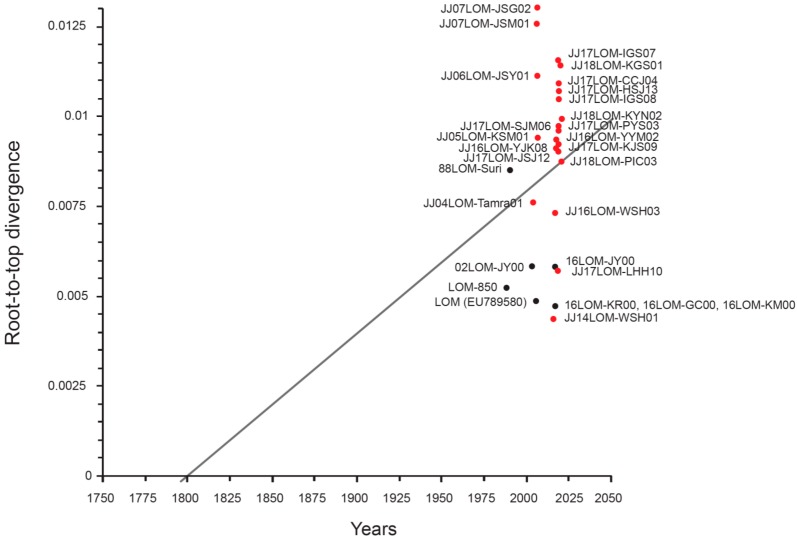
Root-to-top divergence analysis of Jeju LOM strains and commercial LOM vaccine strains. Complete genomes of 29 strains (8 commercial LOM vaccines and 21 Jeju LOM strains) were analyzed using the TempEst v1.5.1 program. The commercial LOM vaccine strains are marked with a black rectangle and the Jeju LOM strains are marked with a red rectangle. The heuristic residual mean squared method used to analyze all strains revealed the following: slope (rate), 3.93 × 10^−5^; X-intercept (TMRCA), 1797.24; correlation coefficient, 0.1272; R squared value, 1.6173 × 10^−2^; and residual mean squared value, 6.9076 × 10^−6^.

**Figure 3 pathogens-08-00251-f003:**
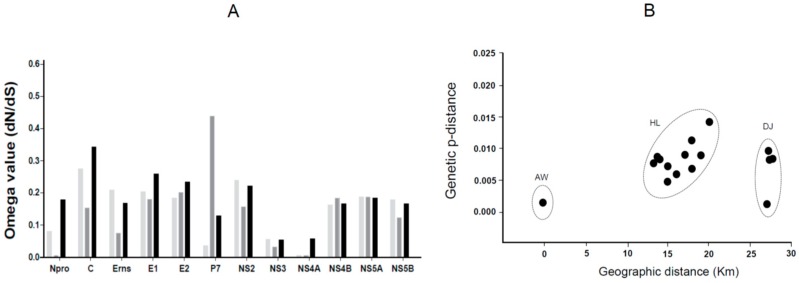
Omega values (*dN/dS*), genetic p-distances, and geographic distances for Jeju LOM strains. The *dN/dS* value for each of the structure proteins (N^pro^, C, E^rns^, E1, E2, p7) and non-structural proteins (NS2, NS3, NS4A, NS4B, NS5A, NS5B) was calculated and compared among 8 commercial LOM vaccine strains, 5 Jeju LOM strains (2004–2007), and 16 Jeju LOM strains (2014–2018) (**A**). Light gray denotes the commercial LOM vaccine strain, dark gray denotes the Jeju LOM strains (2004–2007), and black denotes Jeju LOM strains (2014–2018) (A). Jeju LOM strains (2014–2018) from the Aeweal (AW), Hanlim (HL), and Daejeong (DJ) regions show genetic *p*-distances of 0.0018 (one strain), 0.0045–0.0139 (11 strains) and 0.0011–0.0086 (4 strains), respectively (**B**).

**Figure 4 pathogens-08-00251-f004:**
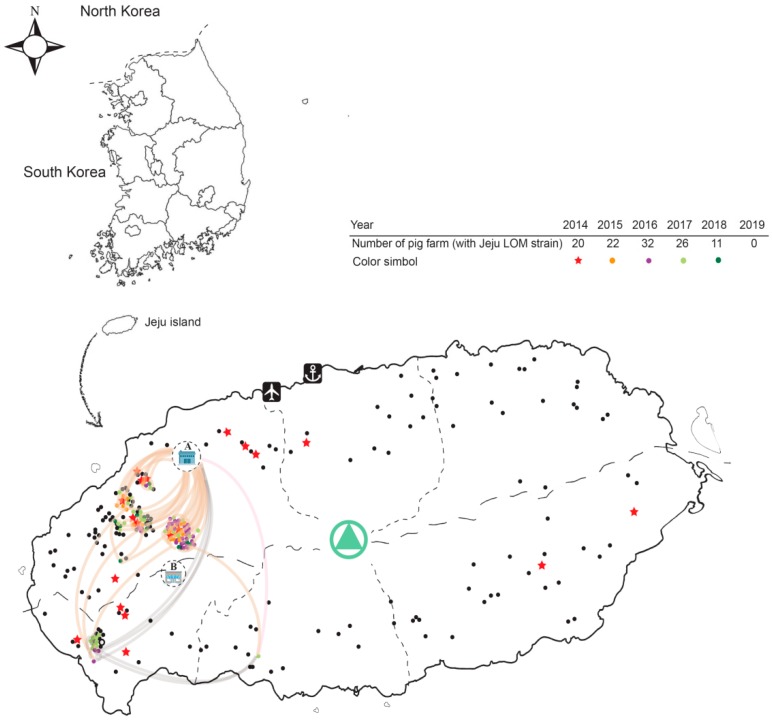
Map of Jeju Island showing locations of the pig farms harboring the Jeju LOM strains (2014–2018). Pig farms harboring Jeju LOM strains (2014–2018) are marked with a red star (2014), an orange rectangle (2015), a purple rectangle (2016), a light green rectangle (2017), or a deep green rectangle (2018). Black rectangles denote pig farms without Jeju LOM strains. An old slaughterhouse (**A**) and a new slaughterhouse (**B**) built at the end of 2018 are located in the upper and middle left, respectively. The light brown and light purple curved lines denote the estimated routes by which Jeju LOM strains entered the pig farms.

**Figure 5 pathogens-08-00251-f005:**
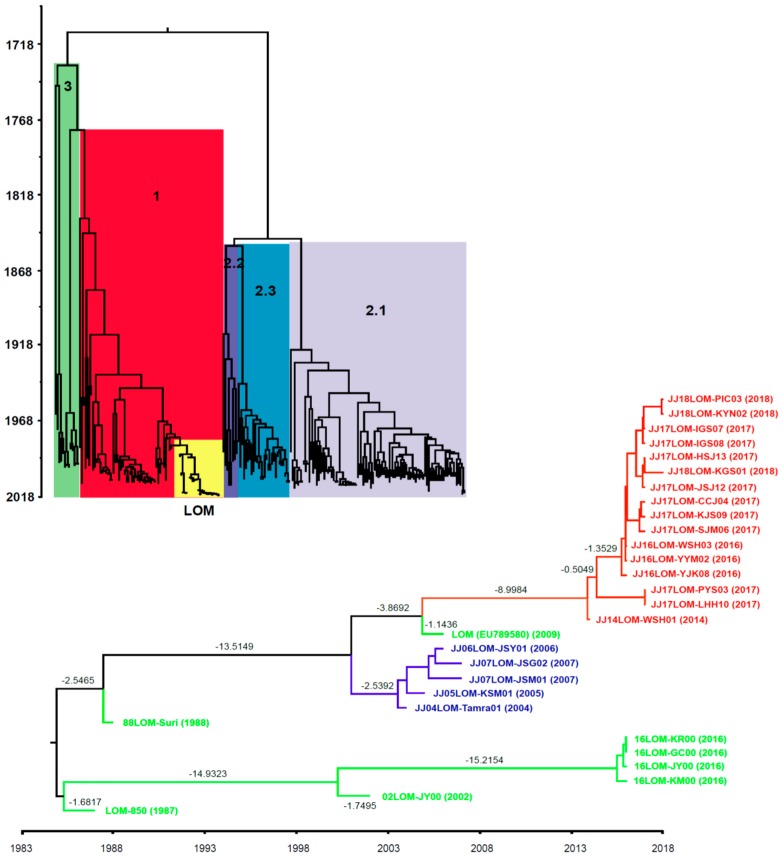
A maximum clade credibility tree based on complete E2 sequences of CSFVs. Rates of nucleotide substitution per site and per year, and the most recent common ancestor (tMRCA), were estimated using a Bayesian MCMC approach. Each dataset was simulated with the following options: generation = 100,000,000, burn-in of 10%, and ESSs > 200. The confidence of the phylogentic analysis is present to numbers representing branch length (time) above the nodes. The yellow block comprises LOM strains within Genotype 1. The light green line denotes 8 commercial LOM vaccine strains. The blue line denotes 5 Jeju LOM strains (2004–2007) and the red line denotes 16 Jeju LOM strains (2014–2018).

**Table 1 pathogens-08-00251-t001:** Classical swine fever virus (CSFV; the Jeju LOM strain) and other pathogens detected in pigs (2014–2018).

	Infection Pattern with CSFV (Jeju LOM Strain)	Pathogens (or Diseases)	Year				
			2014	2015	2016	2017	2018
Suckling piglets	Co-infection with other pathogens	Porcine epidemic diarrhea (PED)			1	2	
		Rotavirus enteritis				2	2
		Rota viral enteritis + *Strep* or *Staphylo* pneumonia			1		1
		*Colibacillosis*				4	2
		*Clostridium difficile* associated disease	2			1	
		*Clostridium* enteritis (+ exudative epidermitis )			1	2(1)	
		Porcine reproductive and respiratory syndrome (PRRS)		2			
		PRRS + *Pasteurella* pneumonia			1		
		*Streptococcal* infection (abscess etc.)			1	3	3
		*Staphylococcal* infection				2	
		Viral encephalitis suspect or bacterial meningitis			1		2
	Infection (only Jeju LOM)				3	6	5
Weaning piglets	Co-infection with other pathogens	PRRS			1		
		PRRS + *Streptococcal* pneumonia			1		
		PRRS + PCV-2 + APP			1		
		*Actinobacillus pleuropneumoniae* (APP)			1	2	
		APP + PCV-2				1	
		PCV-2 (+ *Salmonellosis/Staphylo*)					3
		*Pasteurella* (or *E. coli*) pneumonia			1	1	
		*Streptococcal* infection					4
		Rotavirus enteritis					2
		*Salmonellosis* (*Colibacillosis*)				2(1)	
Total			2	2	14	28(2)	24

**Table 2 pathogens-08-00251-t002:** Immunohistochemical analysis of CSFV (Jeju LOM strain) in the internal organs of pigs.

	Infection Pattern with CSFV (Jeju LOM Strain)	Internal Organs (No. of Positive Pigs/No. Tested)									Total
		Tonsil	Lymph Node	Spleen	Lung	Heart	Kidney	Liver	GI Tract	CNS	
Suckling piglets	Co-infection	11/24	3/30	8/34	4/29	0/32	2/34	1/34	2/33	0/26	17/34
	Infection (only Jeju LOM)	3/11	3/10	3/14	2/12	0/13	0/14	0/13	0/13	0/12	8/14
	Subtotal	14/35	6/40	11/48	6/41	0/45	2/48	1/47	2/46	0/38	25/48
Weaning pigs	Co-infection	1/20	4/20	3/18	3/20	0/20	0/20	0/19	0/21	0/15	6/21
Aborted fetuses	Infection (only Jeju LOM)	NT	NT	0/8	0/8	0/8	1/8	0/8	NT	0/8	1/8

GI: Gastrointestinal, CNS: Central Nervous System.

**Table 3 pathogens-08-00251-t003:** CSF (Jeju LOM) seropositive rates and anti-CSFV (Jeju LOM) antibody titers in pigs of seven farms on Jeju Island.

Pig Farm	Antibody Positive Ratio (%) against CSFV (Jeju LOM Strain)					Average Antibody Titers (Log _2_) for CSFV (Jeju LOM Strain)				
	Sow	Piglet (10–20 days)	Pig (40–60 days)	Pig (90–120 days)	Pig (150–180 days)	Sow	Piglet (10–20 days)	Pig (40–60 days)	Pig (90–120 days)	Pig (150–180 days)
A	80	40	30	0	0	9.5	7.0	9.1	-	-
B	90	95	100	80	0	10.0	10.0	7.1	5.2	-
C	100	100	40	40	90	8.1	6.1	5.5	5.5	8.0
D	100	90	0	10	70	9.1	6.3	-	5.6	8.5
E	90	95	0	0	40	9.5	6.2	-	-	8.9
F	80	65	0	0	10	9.6	6.0	-	-	7.1
G	70	60	0	20	60	10.2	7.7	-	5.8	4.7

**Table 4 pathogens-08-00251-t004:** Contamination by the Jeju LOM strain via exposure during transport to or at a slaughterhouse.

qRT-PCR			Immuno Histo Chemistry (IHC) Staining			
Ct Value (Range)	Jeju LOM Strain * TCID_50_ (Log _10_)	Sample no.	Negative Samples	Weak Positive Samples	Strong Positive Samples	** No Test
>40	<1.0	91	43			48
36.1–39.9	1.0–2.0	26	9	1		16
30.5–36.0	2.0–3.0	21	4	8		9
26.7–30.4	3.0–4.0	92	12	22	3	55
24.4–26.6	4.0–5.0	11		1	9	1
<24.3	>5.0	1			1	

* TCID_50_: Tissue culture infective dose 50. ** No test: samples were inoculated PK-15 cells but IHC was not performed due to contamination by bacteria.

**Table 5 pathogens-08-00251-t005:** Estimated transmission probability between non-inoculated pigs and pigs inoculated with the Jeju LOM strain.

Group	Number of Pigs Inoculated with the Jeju LOM Strain	Non-Inoculated Pigs Exposed to Jeju LOM Virus Inoculated Pigs (1 DPI ^a^)	Pigs Detected with Jeju LOM Strain Antigens and/or Antibodies in Non-Inoculated Pigs (45 DPI)	Transmission Probability Estimate	
				R0 ^b^	95% CI ^c^
1	6	6	2	1.22	0.980–1.765
2	6	6	0	0.00	NA ^d^

R0 calculated as −In((1 − AR)/S0)/(AR − (1 − S0)), CI calculated as AR ± 1.96$\sqrt{\mathrm{ARx}\left(1-\mathrm{AR}\right)/n}$). ^a^ DPI, days post-infection. ^b^ R0: reproduction number; ^c^ CI: confidence interval; ^d^ NA: not applicable.

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
