# Peer review of "Impact of a Live Attenuated Classical Swine Fever Virus Introduced to Jeju Island, a CSF-Free Area"

_pathogens, 2019, doi:10.3390/pathogens8040251_

Round 1

Reviewer 1 Report

The results of this study provide significant concern and awareness about the safety and reliability of CSFV line attenuated strains. Nevertheless, phylogenetic analysis revealed distinct features of CSFV LOM-like viruses. The number of mismatches discovered in those stains allowed to undoubtedly differentiate them from the classical CSFV vaccine strains. In addition, a relatively high transmission rate of CSFV LOM-like strains points out a possible reversion to a virulent phenotype.

The data are of interest and will benefit from more examples of CSFV vaccine strain circulation in the mainland, CSFV LAV-associated pathogenesis? What is the transmission rate for the LAV strains?

The results of the IHC study do not provide any significance to the understanding of the CSFV LOM-like strains pathogenesis. Despite the presence of CSFV antigen, other pathogens may be strongly associated with discovered lesions. 

Minor comments:

Please add some information on the JJ16LOM-YJK08  strain for the transmission study.

Figure 2 - Please add more clarification and details. The legend should be presented. The Boxplot graph looks confusing. The quality of figure is not high enough to see the details and marks.

Table 3 - Please add more explanation. For example, what is the origin of the strong positive sample with TCID50 >5.0?

What does the “No test ( Contamination)” column mean?

Figure 6 - Please add description for the numbers in the phylogenetic tree ( above the branches).

Author Response

Reviewer 1

The results of this study provide significant concern adn awareness about the safety and reliability of CSFV line attenuated strains. Nevertheless, phylogenetic analysis revealed distinct features of CSFV LOM-like viruses. The number of mismatches discovered in those stains allowed to undoubtedly differentiate them from the classical CSFV vaccine strain. In addition, a relatively high transmission rate of CSF LOM-like strains points out a possible reversion to a virulent phenotype. The data are of interest and will benefit from more examples of CSFV vaccine strain circulation in the mainland, CSFV LAV-associated pathogenesis? What is the transmission rate for the LAV strain?

The results of the IHC study do not provide any significance to the understanding of the CSFV LOM-like strains pathogenesis. Despite the presence of CSFV antigen, other pathogens may be strongly associated with discovered lesions.

Answer: For avoiding the confusing for readers, we changed "LOM-like strains" to "Jeju LOM strains" (LOM strains isolated from Jeju pigs). We found to have similar characterization and transmission between LOM vaccine strain using on the mainland and Jeju LOM strains circulating Jeju island.

Minor comments:

Comment 1: Please add some information on the JJ16LOMYJK08 strain for the transmission study.

Answer: We add in M&M 4.4 (Six pigs from each group were inoculated intramuscularly with 2 ml (104.0TCID50/ml) of Jeju LOM strain JJ16LOM-YJK08 which is a representative strain circulating in Jeju between 2014 and 2018).

Comment 2: Figure 2. Please add more clarification and details. The legend should be presented. The Boxplot graph looks confusing. The quality of figure is not high enough to see the details and mark.

Answer: We changed Figure 2 to table 3. And revised in Result 2.2 and M&M 4.2.

Comment 3: Table 3. Please add more explanation. For example, what is the origin of the strong positive sample with TCID50>5.0?.

Answer: We revised in Result 2.3 (A peroxidase-link immuosorbent assay (PLA) virus viability test revealed strong staining of 13 samples (4 driver foot floor, 2 vehicle wheels and 7 pig-holding compartment) and weak staining of 32 samples; 129 samples were unconfirmed due to cellular contamination by bacteria (Table 4).

Comment 4: What does the "No test (Contamination)" column mean?

Answer: We revised in Result 2.3 (129 samples were unconfirmed due to cellular contamination by bacteria (Table 4). And we add in Table 3 " ** No test: samples were inoculated PK-15 cell but IHC was not performed due to contamination by bacteria."

Comment 5: Figure 6. Please add description for the numbers in the phylogeneic tree (above the branches).

Answer: We change Figure 6 to Figure 5. And we add the explain for number above the branches in Figure 5 regend (The confidence of the phylogentic analysis is present to numbers represent Branch length (time) above the nodes).

Reviewer 2 Report

In general, a well-written paper, with a sufficient amount of novel experimental data. However key passages have to be reconsidered. The abstract needs some reconsideration. It is important to be very precise in the description of LOM and LOM-like viral strains, as this may become confusing for readers.

The final conclusions in the last paragraph of the discussion are not based on data presented in the paper. Therefore, some textual changes should be made here.

Abstract:

line 21: 'in combination with' instead of 'and'

line 24: reconsider phrasing 'were not transmitted', you mean that healthy pigs were not infected by pigs harboring the LOM-like strain

line 25-26:

I think you mean that you could not identify mutations previously identified in the 2004-2007 LOM-like strains. 

line 27: starting from However, is confusing sentence

I suggest you mention that LOM-like strains of 2004-2007 are genetically distinghuisable from 2014-2018, with different mutations, and potential differences in pathogenicity and spread.

line 60: comma after 'sows

line 62: has caused

Line 69: you mention here that the outbreaks of 2004-2008 are from a LOM-strain: this is confusing because in the rest of the paper these outbreaks are referred to as outbreaks of a LOM-like strain

line 76: accidently: the pigs were inoculated with a vaccine that was accidently contaminated with LOM-like strains?

Line 78: LOM-like?

Figure 2 does not contain enough contrast for publication and shoud be re-designed.

Table 3: also mention in a legend to the table that TCID50 values are estimates.

Experiment presented in Table 4:

The conclusion you reach here is based on a very small number of animals. This is understandable because one cannot design an experiment with large group sizes, but please clarify why this experiment should be regared as being representative for what happens in the field. Did you perform calculations on group size?

Figure 3: again, not readable

Line 214: I do not understand the word contaminating. The contamination was in the ERY-ALC vaccine, with a LOM-like strain, if I understand your paper correctly. Again, please be precise in discriminating between commercial LOM-strains (=vaccine strains as mentioned M&M) and LOM-like sequences that accidently have been introduced on the island, most probably after having circulated on the mainland (and attenuation/mutation).

Line 245

The main VIRAL characteristic of the LOM strain are attenuated compared to virulent CSF

Discussion paragraph 1:

The comparison between R-values of the healthy and unhealthy group should be put in perspective. Is your group size approprate to draw conclusions? The suggestion is made that virus shedding is prolongued in the 'unhealthy' group, but as far a I can see this was not experimentally addressed. The statement is based on literature. In the same paragraph, discussion is switched to the operation of a new slaughter house. The subject of spread of the LOM-like virus because of repeated visits of contaminated vehicles should not be part of a discussion on differences in transmission between pigs.

I think the discussion should focus on differences between LOM, LOM-like strains 2004-2008 and LOM-like strains 2014-2018 as described in the paper. You identify novel mutations, and can discuss potential relation to virulence, pathogenicity etc. The conclusions reached in line 294-298 are based on experimental data in the paper and should be emphasized.

Line 299 onwards, which start with 'in conclusion', contains a list of factors that indeed may be related to infection on pig farms, but these factors are not immediately correlated to the data that you present in the paper. The section does not contain an extrapolation of the new experimental data into risk factors.

Author Response

Reviewer 2

In general, a well-written paper, with a sufficient amount of novel experimental data. However key passages have to be reconsidered. The abstract needs some reconsideration. It is important to be very precise in the description of LOM and LOM-like viral strains, as this may become confusing for readers.

Answer: For avoiding the confusing for readers, we changed "LOM-like strains" to "Jeju LOM strains" (LOM strains isolated from Jeju pigs).

The final conclusions in the last paragraph of the discussion are not based on data presented in the paper. Therefore, some textual changes should be made here.

Answer: According to comment of reviewer, we revised the final conclusions part at a new revised manuscript.

Abstract:

Comment 1: line 21: 'in combination with' instead of 'and'

Answer: We change "and" to "in combination with" in abstract.

Comment 2: line 24: reconsider phrasing 'were not transmitted', you mean that healthy pigs were not infected by pigs harboring the LOM-like strain

Answer: We revised to a new sentence in abstract.

Comment 3: line 25-26:I think you mean that you could not identify mutations previously identified in the 2004-2007 LOM-like strains. 

Answer: We revised to a new sentence in abstract.

Comment 4: line 27: starting from However, is confusing sentence

I suggest you mention that LOM-like strains of 2004-2007 are genetically distinghuisable from 2014-2018, with different mutations, and potential differences in pathogenicity and spread.

Answer: We revised to a new sentence in abstract.

Comment 5: line 60: comma after 'sows

Answer: We added comma after "sows" in introduction.

Comment 6: line 62: has caused

Answer: We revised "have" to "has" in introduction.

Comment 7:Line 69: you mention here that the outbreaks of 2004-2008 are from a LOM-strain: this is confusing because in the rest of the paper these outbreaks are referred to as outbreaks of a LOM-like strain

Answer: We revised to Jeju LOM strains. Jeju LOM strains (2004-2007) isolated from Jeju pigs from 2004 to 2007. Jeju LOM strains (2014-2018) isolated from Jeju pigs from 2014 to 2018.

Comment 8: line 76: accidently: the pigs were inoculated with a vaccine that was accidently contaminated with LOM-like strains?

Answer: For avoiding the confusing for readers, we changed LOM-like strains to Jeju LOM strains (LOM strains isolated from Jeju pigs). And according to LOM isolation years from pigs in Jeju island, we named as Jeju LOM strains (2004-2007) and Jeju LOM strains (2014-2018).

Comment 9: Line 78: LOM-like?

Answer: We revised LOM strain to LOM vaccine strain in introduction.

Comment 10: Figure 2 does not contain enough contrast for publication and should be re-designed.

Answer: We changed Figure 2 to table 3. And revised in Result 2.2 and M&M 4.2.

Comment 11: Table 3: also mention in a legend to the table that TCID50 values are estimates.

Answer: We added the explain for TCID50 in Table 3.

Comment 12: Experiment presented in Table 4:

The conclusion you reach here is based on a very small number of animals. This is understandable because one cannot design an experiment with large group sizes, but please clarify why this experiment should be regared as being representative for what happens in the field. Did you perform calculations on group size?

Answer: We are enough understand the reviewer's comment that small number experiment of animals is difficult representative in the field status. We experimented base on CSFV OIE manual (Manual of standards for diagnostic tests and vaccines). Although we tested small number of pigs, we can confirmed possibility of the transmission pig-to-pig because we had repeated test two times. And we revised the sentence in discussion.

Comment 13: Figure 3: again, not readable

Answer: We change Figure 3 to Figure 2 in new revised manuscript. According to reviewer's comment, we revised the figure 2 regend.

Comment 14: Line 214: I do not understand the word contaminating. The contamination was in the ERY-ALC vaccine, with a LOM-like strain, if I understand your paper correctly. Again, please be precise in discriminating between commercial LOM-strains (=vaccine strains as mentioned M&M) and LOM-like sequences that accidently have been introduced on the island, most probably after having circulated on the mainland (and attenuation/mutation).

Answer: For avoiding the confusing for readers, we changed LOM-like strains to Jeju LOM strains (LOM strains isolated from Jeju pigs). And according to LOM isolation years from pigs in Jeju island, we named as Jeju LOM strains (2004-2007) and Jeju LOM strains (2014-2018). And commercial LOM-strains changed to commercial LOM vaccine strain.

Comment 15: Line 245

The main VIRAL characteristic of the LOM strain are attenuated compared to virulent CSF

Discussion paragraph 1:

The comparison between R-values of the healthy and unhealthy group should be put in perspective. Is your group size approprate to draw conclusions? The suggestion is made that virus shedding is prolongued in the 'unhealthy' group, but as far a I can see this was not experimentally addressed. The statement is based on literature. In the same paragraph, discussion is switched to the operation of a new slaughter house. The subject of spread of the LOM-like virus because of repeated visits of contaminated vehicles should not be part of a discussion on differences in transmission between pigs.

I think the discussion should focus on differences between LOM, LOM-like strains 2004-2008 and LOM-like strains 2014-2018 as described in the paper. You identify novel mutations, and can discuss potential relation to virulence, pathogenicity etc. The conclusions reached in line 294-298 are based on experimental data in the paper and should be emphasized.

Answer: According to reviewer's comment, we removed and revised the sentences in discussion.

Comment 16: Line 299 onwards, which start with 'in conclusion', contains a list of factors that indeed may be related to infection on pig farms, but these factors are not immediately correlated to the data that you present in the paper. The section does not contain an extrapolation of the new experimental data into risk factors.

Answer: We revised in conclusion that " In conclusion, CSF-like histopathogenic lesion of Jeju pigs revealed to be more related with other viral pathogens rather than Jeju LOM strains (2014-2018), despite the presence of Jeju LOM strain in organs of piglets. We also confirmed the pig-to-pig transmission of Jeju LOM strain and farm-to-farm transmission may be caused by vehicles visiting the slaughterhouse. Although we found some genetic differences between Jeju LOM strains (2014-2018) and commercial LOM vaccine strains, the more pathogenesis studies may be needed using animals such as pregnant sows and SPF pigs."

Reviewer 3 Report

Classical swine fever virus is a cause of an acute, highly infectious and economically damaging disease in swine. As the vaccine again CSFV is present, vaccination is used to prevent the number of CSF outbreaks worldwide. The manuscript entitled: “Impact of a live attenuated classical swine fever virus introduced onto Jeju Island, a CSF-free area”, by Choe et al., describes the investigation of pig-to-pig transmission and the possibility of reversion-to-pathogenicity of LOM-like strain viruses. The manuscript is well written, and the experiments are technically adequately done. However, some issues identified at both, experimental and interpretation levels need to be addressed to clarify the study:

Section 2.1. – it is not clear whether the 122 samples were tested for the presence of CSFV in this study or in the previous one. Most of the Figures presented in the paper are illegible, in particular Figure 2 and 4. Section 2.3. – What is the idea of PLA assay? It is not explained either in Results nor in Material and Methods section. Line 146 (section 2.3) – 129 samples were unconfirmed due to the contamination. What contaminations were present? How is it possible as 151 form 242 samples were confirmed positive for CSFV in qRT-PCR? Does it mean that qRT-PCR was performed in contaminated probes? Section 4.1. – Were the samples tested for CSFV, PCV2, PRRSV etc. before or during this study? If before what was done according to this study? Section 4.2. – NPLA assay should be described more in details or a citation should be added. Section 4.3. (line 338 and 339) – Again were the samples tested in this or previous study? If in this study, then these sentences about positive samples should be deleted and present only in Results section. Section 4.3. - PLA assay should be described more in details or a citation should be added.

Author Response

Reviewer 3

Classical swine fever virus is a cause of an acute, highly infectious and economically damaging disease in swine. As the vaccine again CSFV is present, vaccination is used to prevent the number of CSF outbreaks worldwide. The manuscript entitled: “Impact of a live attenuated classical swine fever virus introduced onto Jeju Island, a CSF-free area”, by Choe et al., describes the investigation of pig-to-pig transmission and the possibility of reversion-to-pathogenicity of LOM-like strain viruses. The manuscript is well written, and the experiments are technically adequately done. However, some issues identified at both, experimental and interpretation levels need to be addressed to clarify the study:

Comment 1: Section 2.1. – it is not clear whether the 122 samples were tested for the presence of CSFV in this study or in the previous one.

Most of the Figures presented in the paper are illegible, in particular Figure 2 and 4.

Answer: The122 samples were tested for the presence of CSFV in this study. Figure 2 changed to Table 3 and figure 4 changed to figure 3. Figures legends revised.  

Comment 2: Section 2.3. – What is the idea of PLA assay? It is not explained either in Results nor in Material and Methods section.

Answer: We added the PLA assay method in M&M 4.3.

Comment 3:Line 146 (section 2.3) – 129 samples were unconfirmed due to the contamination. What contaminations were present? How is it possible as 151 form 242 samples were confirmed positive for CSFV in qRT-PCR? Does it mean that qRT-PCR was performed in contaminated probes?

Answer: We revised in Result 2.3 (129 samples were unconfirmed due to cellular contamination by bacteria (Table 4). And we add in Table 3 " ** No test: samples were inoculated PK-15 cell but IHC was not performed due to contamination by bacteria."

Comment 4: Section 4.1. – Were the samples tested for CSFV, PCV2, PRRSV etc. before or during this study? If before what was done according to this study?

Answer: All experiments were performed this study.

Comment 5: Section 4.2. – NPLA assay should be described more in details or a citation should be added.

Answer: We added the NPLA assay method in M&M (section 4.2).

Comment 6: Section 4.3. (line 338 and 339) – Again were the samples tested in this or previous study? If in this study, then these sentences about positive samples should be deleted and present only in Results section.

Answer: All experiments were performed this study. According to reviewer's comment, we revised M&M (section 4.3).

Comment 7: Section 4.3. - PLA assay should be described more in details or a citation should be added.

Answer: We added the PLA assay method in M&M (section 4.3).

Round 2

Reviewer 1 Report

The manuscript was significantly improved and all the reviewer's comments were satisfactory addresses.  

Minor changes:

I would suggest to use the word "into" instead of "onto" in the title.   Line 30 in the abstract, please change "were" to "was", or use plural form for "histopathogenic investigation".  Line 32 in the abstract,  please add a comma, after the word "pathogens".  Line 35 in the abstract, please change "transmission of Jeju LOM strains may be carried out" to "transmission factor for Jeju LOM strains circulation is ..." Line 40, use a comma after the word "however". Use "whether" before "determine" Line 41, use "were" instead of "are". 

Author Response

Reviewer 1

This manuscript was significantly improved and all the reviewer's comments were satisfactory addresses

Minor comments:

Comment 1: I would suggest to use the word "into " instead of " onto" in the tile.

Answer: We revised to "into" the title of the revised manuscript.

Comment 2: Line 30 in the abstract, please change "were" to "was", or use plural form for "histopathogenic investigation".

Answer: We changed were to was in the abstract.

Comment 3: Line 32 in the abstract, please add a comma, after the word "pathogens" Answer: We add comma after the word "pathogens" in the abstract.

Comment 4: Line 35 in the abstract, please change "transmission of Jeju LOM strains may be carried out" to "transmission factor for Jeju LOM strains circulation is..."

Answer: We changed to "transmission factor for Jeju LOM strains circulation is..." in the abstract.

Comment 5: Line 40, use a comma after the word "however". Use "whether" before "determine"

Answer: We add a comma after the word "however". And we used "whether" before "determine".

Comment 6: Line 41, use "were" instead of "are"

Answer: We change "are" to "were" in abstract.

Reviewer 3 Report

Most of the required corrections have been done and explanations have been provided. In my opinion this revised version of the manuscript will be suitable for publication after two minor corrections:

In the previous review I suggested to correct most of Figures due to their illegibility (especially Figure 2 and 4). In the revised version of the paper no Figures are included so it is impossible to verify. Section 4.3. PLA assay was used to detect Jeju LOM strains not to isolate them.

Author Response

Reviewer 3

Most of the required corrections have been done and explanations have been provided. In my opinion this revised version of the manuscript will be suitable for publication after two minor correction.

Comment 1: In the previous review I suggested to correct most of Figures due to their illegibility (especially Fig 2 and 4). In revised version of the paper no Figures are included so it is impossible to verify.

Answer: Original figure 2 changed to Table 3 in revised manuscript. And we also change original figure 5 to 4 and change original figure 6 to 5 in revised manuscript. And tables change original table 3 to 4 and original table 4 to 5 in revised manuscript. We contained all tables and all figures in revised manuscript. Table 3 (original figure 2) will be easier to understand than the original figure 2. We revised the figure regend to make it easier to understand the figure 3 (original figure 4).

Comment 2: Section 4.3. - PLA assay used to detect Jeju LOM strains not to isolate them.

Answer: According to reviewer's comment. we revised to "A peroxidase-linked Immunosorbent assay (PLA) was used to confirm viability of Jeju LOM strains." in section 4.3
